

# Multi-feature fusion and dandelion optimizer based model for automatically diagnosing the gastrointestinal diseases

Soner Kiziloluk[1], Muhammed Yildirim[1], Harun Bingol[2] and Bilal Alatas[3]

[1] Computer Engineering, Malatya Turgut Ozal University, Malatya, Turkey
[2] Software Engineering, Malatya Turgut Ozal University, Malatya, Turkey
[3] Software Engineering, Firat (Euphrates) University, Elazig, Turkey

## ABSTRACT

It is a known fact that gastrointestinal diseases are extremely common among the public. The most common of these diseases are gastritis, reflux, and dyspepsia. Since the symptoms of these diseases are similar, diagnosis can often be confused. Therefore, it is of great importance to make these diagnoses faster and more accurate by using computer-aided systems. Therefore, in this article, a new artificial intelligence-based hybrid method was developed to classify images with high accuracy of anatomical landmarks that cause gastrointestinal diseases, pathological findings and polyps removed during endoscopy, which usually cause cancer. In the proposed method, firstly trained InceptionV3 and MobileNetV2 architectures are used and feature extraction is performed with these two architectures. Then, the features obtained from InceptionV3 and MobileNetV2 architectures are merged. Thanks to this merging process, different features belonging to the same images were brought together. However, these features contain irrelevant and redundant features that may have a negative impact on classification performance. Therefore, Dandelion Optimizer (DO), one of the most recent metaheuristic optimization algorithms, was used as a feature selector to select the appropriate features to improve the classification performance and support vector machine (SVM) was used as a classifier. In the experimental study, the proposed method was also compared with different convolutional neural network (CNN) models and it was found that the proposed method achieved better results. The accuracy value obtained in the proposed model is 93.88%.

# INTRODUCTION

The gastrointestinal system has a very complex structure. The complexity of this system stems from the fact that it contains many different regions such as the mouth, esophagus, stomach, small and large intestine (*Modi et al., 2023*; *Peery et al., 2012*). Gastrointestinal diseases are extremely important and affect the comfort of human life. It involves examining and evaluating both single and multiple images together for the diagnosis of some diseases. For this reason, it is not easy to diagnose this type of disease by looking at a

Corresponding author
Bilal Alatas, balatas@firat.edu.tr

single image. The disease is diagnosed by specialist physicians. However, the difficulty in diagnosing gastrointestinal diseases may arise from the fact that even medical experts make mistakes when examining endoscopy images or cannot make a clear distinction and classify the image correctly. In such cases, computer-aided image classification techniques are frequently used (*Yildirim et al., 2023a*; *Bingol et al., 2023*; *Yildirim et al., 2023b*; *Bugday et al., 2023*). It is a known fact that artificial intelligence techniques can detect many different diseases quite successfully. The success of artificial intelligence techniques often occurs by extracting valuable features of the image and using these features in classification (*Jiang et al., 2017*). When dealing with real world problems, it is extremely difficult to present a mathematical model of the problem. Classifying images used in disease diagnosis is also a real-world problem.

In this study, the features of the disease-causing images are extracted using InceptionV3 (*Szegedy et al., 2016*) and MobileNetV2 (*Sandler et al., 2018*) convolutional neural network (CNN) architectures, which are frequently used in the literature. The feature maps obtained with MobileNetV2 and InceptionV3 architectures are then concatenated. The features generated by CNN architectures usually come with some complexities in terms of size. These features include redundant and noisy features that have a negative impact on the classification model accuracy. Therefore, feature selection should be performed aiming to improve classification performance by selecting important features. The feature selection process is an NP-hard combinatorial problem. Since classical mathematical methods cannot handle solving NP-hard problems, it is a good way to use metaheuristic algorithms in this type of problem. Because metaheuristic algorithms are designed to find near-optimal solutions to high-dimensional complex problems. The Dandelion Optimizer (DO) algorithm proposed by *Zhao et al. (2022)* is a novel metaheuristic optimization algorithm inspired by the dandelion plants reproduction through its seeds scattered in the wind. The success of a metaheuristic optimization algorithm is highly dependent on the balance of the exploration and exploitation steps of the algorithm. The Rising and Landing steps of the DO ensure that the algorithm's exploration and exploitation are balanced. Furthermore, in *Zhao et al. (2022)*, tested the DO on 30 different mathematical optimization test problems and four different engineering design problems. The experimental results show that the DO outperforms nine well-known metaheuristic algorithms namely the whale optimization algorithm, the sine cosine algorithm, the horse herd optimization algorithm, the Levy flight distribution, the chimp optimization algorithm, the Aquila optimizer, the seagull optimization algorithm, the Harris hawks optimization, and the moth swarm algorithm. These reasons mentioned above are the motivation for choosing the DO algorithm for feature selection in this study. The features selected by the DO algorithm process are classified in the support vector machine (SVM) (*Burges, 1998*) classifier. This selection process is considered as an optimization problem and the feature map is also considered as a search space. Therefore, in this article, classifying gastrointestinal diseases is considered a real-world problem and tried to be solved.

## Related works

Automatic detection of gastrointestinal diseases by computer-aided systems will allow both the early start of the treatment process and the lightening of the workload of specialists. Therefore, there are some studies in the literature to detect gastrointestinal diseases. *Lonseko et al. (2021)* proposed a deep learning-based model that classifies gastrointestinal diseases using endoscopic images. They stated that in their study, they compared the models they proposed with state-of-the-art models and achieved a better accuracy value than them with 93.19%. However, the authors stated that they used data augmentation methods to eliminate the data imbalance problem (*Lonseko et al., 2021*).

*Nouman Noor et al. (2023)* proposed a hybrid model to classify gastrointestinal diseases. The authors classified the new dataset they obtained by applying image enhancement techniques to the original datasets using the MobileNetV2 architecture. In their proposed model, they increased the total number of 4,854 original images to 30,000 images by combining two different datasets and using data augmentation techniques. The authors state that they reached an accuracy value of 96.40%. However, data augmentation techniques are undesirable as they may cause the model to over-fitting and also prolong the training time of the model (*Nouman Noor et al., 2023*).

*Mohapatra et al. (2021)* tried to classify the Kvasir dataset consisting of 8,000 endoscopic images with the Discrete Wavelet Transform and CNN-based model they proposed. Data augmentation was carried out by applying rotation to the images in the original dataset, and the number of images in the dataset was increased to 28,800, while the accuracy rate of the proposed model was stated to be 97.25%. However, it is known that data multiplexing techniques may cause the model to memorize (*Mohapatra et al., 2021*).

In their study, *Sali et al. (2020)* mentioned that they placed a class in the VGGNet architecture and thus benefited from the hierarchical structure of the layers. They stated that the hierarchical model they proposed was more successful in classifying gastrointestinal diseases than the flat model VGGNet. It was stated that the obtained classification accuracy values varied between 93.7% and 98.7% (*Sali et al., 2020*).

*Sharif et al. (2021)* stated that in order to classify gastrointestinal tract diseases, geometric features are selected from the diseased region by performing lesion segmentation and sent to the KNN classifier. They stated that 5,500 wireless capsule endoscopic images were used during the experiments and the best classification accuracy was 99.42% (*Sharif et al., 2021*).

*Varalaxmi et al. (2023)* stated that traditional methods are time-consuming in the classification of gastrointestinal diseases, and instead they used the ResNet50 architecture, which is extremely accurate and does not cause loss of time, thus achieving an accuracy of 88.05% (*Varalaxmi et al., 2023*).

*Montalbo (2022)* stated that in his study for the detection of gastrointestinal diseases, he developed a CNN-based model that reduces the processing load. The developed model was expressed in three stages by the researcher. The developed model is a multi-fused model with auxiliary layers, alpha outputs and fusion residue block. In this study conducted for

the detection of gastrointestinal diseases, an accuracy value of 97.25% was achieved after data preprocessing steps were applied to the dataset. While achieving a high accuracy value in the proposed method is the main advantage of the study, training a model from scratch can be shown as a disadvantage of the study (*Montalbo, 2022*).

### Contribution and novelty

- The high similarity of symptoms that distinguish gastrointestinal diseases from each other makes the diagnosis process difficult and this negatively affects the beginning of treatment. Therefore, it is very important to detect gastrointestinal diseases with computer-aid systems.
- In this study, a hybrid model was created for the detection of gastrointestinal disorders. In the developed model, feature extraction was first carried out with InceptionV3 and MobileNetV2 pre-trained models, which are accepted in the literature, in order for the Dandelion Optimizer convolutional neural network (DO-CNN) to work faster.
- With InceptionV3 and MobileNetV2 architectures, different features of the same images are extracted and these features are combined. In this way, the performance of the proposed model is increased.
- The use of DO, a novel and highly effective metaheuristic technique, to reduce the size of the obtained features is one of its significant contributions. In this way, the size of the acquired features is decreased using the DO approach. Features that are superfluous are removed in this way.
- Finally, the feature map optimized with the DO method was classified in the SVM classifier and an accuracy value of 93.88% was obtained.

### Organization of article

In the rest of the article, the dataset used, the DO algorithm, and the proposed DO-CNN model are explained in the Material and Methods section. Comparative experimental results are presented in the Results section. Finally, the Discussion and Conclusion sections are presented.

## MATERIALS AND METHODS

In this section, the dataset used in the article, the metaheuristic Dandelion algorithm and the DO-CNN are detailed. The rough flow chart of the DO-CNN is shown in Fig. 1.

### Kvasir dataset

The Kvasir dataset, consisting of a total of 4,000 endoscopic images and eight classes, with an equal number of images in each class, was used in the classification of gastrointestinal diseases. The dataset is divided into three groups: anatomical landmarks, pathological findings and cancer-causing polyps. The first group, anatomical landmarks, includes z-line, pylorus and cecum. The second group includes esophagitis, polyps and ulcerative colitis. In the last group, there are dyed and lifted polyps and dyed resection margins (*Pogorelov et al., 2017*).

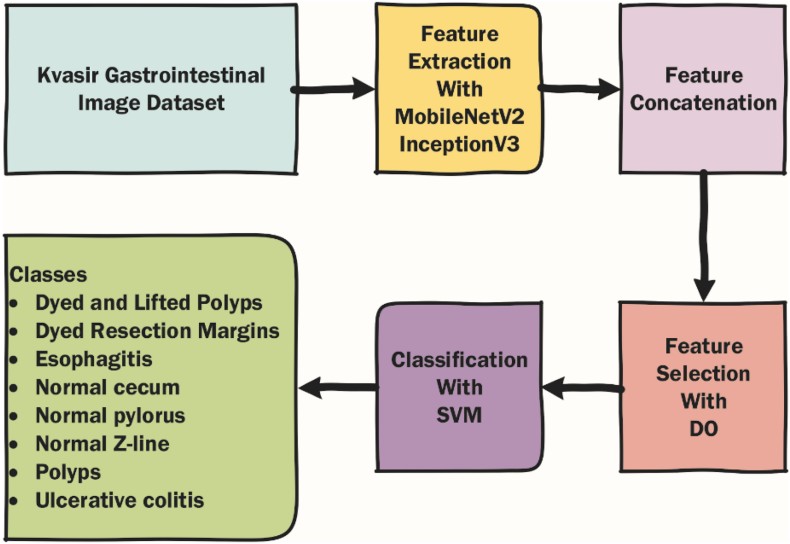

**Figure 1  Rough flow chart of the DO-CNN.**

## Dandelion Optimizer

The Dandelion Optimizer (DO), proposed by *Zhao et al. (2022)*, is a novel metaheuristic optimization algorithm inspired by the dandelion plant's reproduction through its seeds scattered in the wind. DO has three phases: Rising, Descending and Landing.

DO starts with an initial population that is randomly generated within the bounds of the search space. Each element in the initial population is a candidate solution and each candidate solution is called a seed. The initial population is created based on Eq. (1).

$$S_i = rnd \times (U - L) + L \tag{1}$$

In the equation, $Si$ is the $i$th seed in the population, $rnd$ is an random value between 0 and 1, $U$ and $L$ are the lower and upper bounds of the search space (*Zhao et al., 2022*).

### Rising stage

This stage is divided into two sub-stages according to the weather: clear day and rainy day (*Zhao et al., 2022*).

**Sub-stage 1:** In clear day, the seeds, influenced by the wind speed, rise in a spiral and have a greater chance of traveling farther along the Y-axis according to the wind speed. In this way, exploration is realized (*Zhao et al., 2022*). This sub-stage is expressed mathematically using Eq. (2).

$$S_{t+1} = S_t + \alpha \times v_x \times v_y \times lnY \times (S_S - S_t) \tag{2}$$

In Eq. (2), $S_t$ is the position of the seed at the $t$-th iteration and $S_s$ is a randomly chosen position as in the Eq. (3). $S_s$ is calculated as in the Eq. (3).

$$S_S = rnd(1, Dim) \times (U - L) + L \tag{3}$$

$lnY$ is the lognormal distribution as shown in Eq. (4).

$$lnY = \begin{cases} \dfrac{1}{y\sqrt{2\pi}} exp\left[-\dfrac{1}{2\sigma^2}(\ln y)^2\right], & y \geq 0 \\ 0, & y < 0 \end{cases} \tag{4}$$

In Eq. (4), $y$ is $N(0, 1)$ standard normal distribution and $\sigma^2 = 1$. $\alpha$ in Eq. (2) is used to determine the step length and is calculated using Eq. (5)

$$\alpha = rnd \times \left(\frac{1}{T^2}t^2 - \frac{2}{T}t + 1\right) \tag{5}$$

where $T$ is the maximum number of iterations and $t$ is the current iteration. $v_x$ and $v_y$ in Eq. (2) are calculated as in Eqs. (6)–(8).

$$v_x = r \times cos\theta \tag{6}$$
$$v_y = r \times sin\theta \tag{7}$$
$$r = \frac{1}{e^\theta} \tag{8}$$

$\theta$ is an arbitrary value between $-\pi$ and $\pi$ (*Zhao et al., 2022*).

**Sub-stage 2:** In rainy day, seeds cannot rise due to moisture, air resistance, *etc*. Therefore, they reach closer locations, which leads to exploitation. Mathematical representation of this situation is represented in Eq. (9).

$$S_{t+1} = S_t \times k \tag{9}$$

In Eq. (9), $k$ is used to regularize the local search and is calculated as in Eqs. (10) and (11).

$$k = 1 - rnd \times q \tag{10}$$
$$q = \frac{1}{T^2 - 2T + 1}t^2 - \frac{2}{T^2 - 2T + 1} + 1 + \frac{1}{T^2 - 2T + 1} \tag{11}$$

The parameter $k$ allows for longer steps in the first iterations and shorter steps towards the last iterations. In other words, the parameter $k$ gradually approaches 1 in order to guarantee convergence to the optimal seed in the last iteration (*Zhao et al., 2022*). As a result, the general mathematical expression of the rising stage is represented by Eq. (12). In the Eq. (12), *rndn* is a normally distributed random number. It is used to dynamically control exploitation and exploration. To make the algorithm more global search-oriented, the breakpoint is set to 1.5. This setting ensures that the dandelion seeds traverse the entire search space as much as possible in the first stage to provide the right direction for the next stage of iterative optimization (*Zhao et al., 2022*).

$$S_{t+1} = \begin{cases} S_t + \alpha \times v_x \times v_y \times lnY \times (S_S - S_t), & rndn < 1.5 \\ S_t \times k, & else \end{cases} \tag{12}$$

### Descending stage

In the descending stage, the seeds begin to descend regularly and this movement is carried out by brown motion. Since the brown motion obeys a normal distribution at each change, it is easy for the seeds to move around more locations in the iterative update process. In addition, the average position information of all seeds is used during descent, which makes it easier for the population to get closer to the optimal solution. At this stage, the proposed DO algorithm also emphasizes exploration. This stage is formulated by the Eq. (13).

$$S_{t+1} = S_t - \alpha \times \beta_t \times (S_{mean_t} - \alpha \times \beta_t \times S_t) \tag{13}$$

In Eq. (13), $\beta_t$ is an arbitrary number from the standard normal distribution, denoting Brownian motion. $S_{mean_t}$ is the average position of the seeds in the population and is calculated using Eq. (14) (*Zhao et al., 2022*).

$$S_{mean_t} = \frac{1}{pop} \sum_{i=1}^{pop} S_i \tag{14}$$

### Landing stage

In the landing stage, the algorithm emphasizes exploitation. In this stage, seeds land in random locations based on the first two stages. To more accurately converge to the global optimum, the seeds use the information of the current elite seed (*Zhao et al., 2022*). This step is mathematically represented using Eq. (15):

$$S_{t+1} = S_{elite} + levy(\lambda) \times \alpha \times (S_{elite} - S_t \times \delta) \tag{15}$$

Here $S_{elite}$ represents the position of the seed with the best fitness value at iteration $i$. *levy* ($\lambda$) is levy flight and is calculated using Eq. (16).

$$levy(\lambda) = s \times \frac{w \times \sigma}{|t|^{\frac{1}{\beta}}} \tag{16}$$

Here $\beta$ is an arbitrary number between 0 and 2, and $s = 0.01$. $w$ and $t$ are arbitrary numbers between 0 and 1. $\sigma$ in Eq. (16) is calculated according to Eq. (17).

$$\sigma = \left[ \frac{\Gamma(1 + \beta) \times \sin\left(\frac{\pi \beta}{2}\right)}{\Gamma\left(\frac{1 + \beta}{2}\right) \times \beta \times 2^{\left(\frac{\beta - 1}{2}\right)}} \right] \tag{17}$$

In Eq. (17), $\beta$ is fixed at 1.5. $\delta$ is a linearly increasing value between 0 and 2 and is calculated based on Eq. (18).

$$\delta = \frac{2t}{T} \tag{18}$$

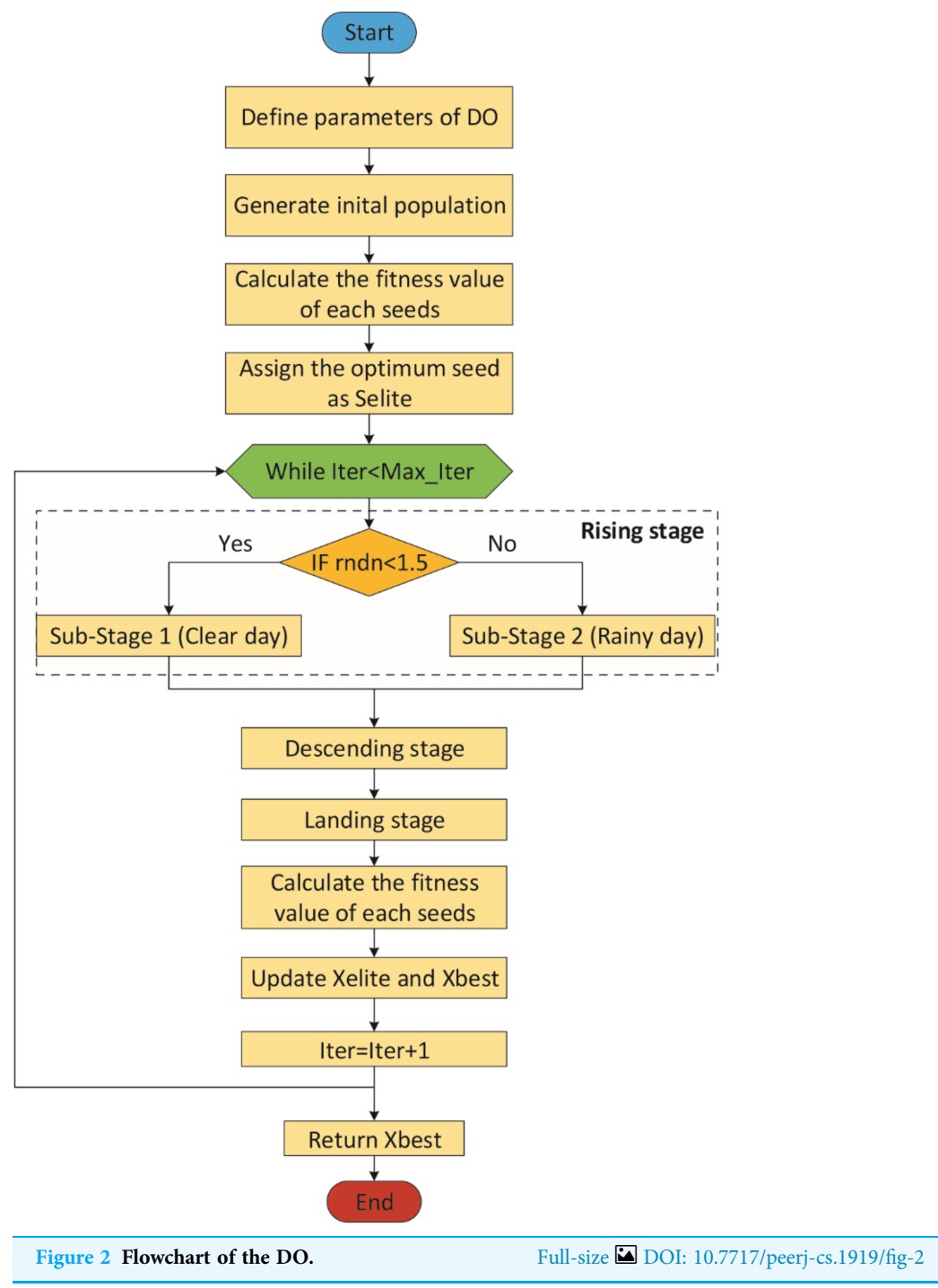

**Figure 2  Flowchart of the DO.**               

Figure 2 shows the flowchart of the DO (*Zhao et al., 2022*).

## Proposed model (DO-CNN)

A DO-CNN model was improved in this study for the classification of gastrointestinal diseases. To compare the performance of the developed model, AlexNet (*Krizhevsky, Sutskever & Hinton, 2012*), ResNet50 (*He et al., 2016*), InceptionV3 (*Szegedy et al., 2016*),

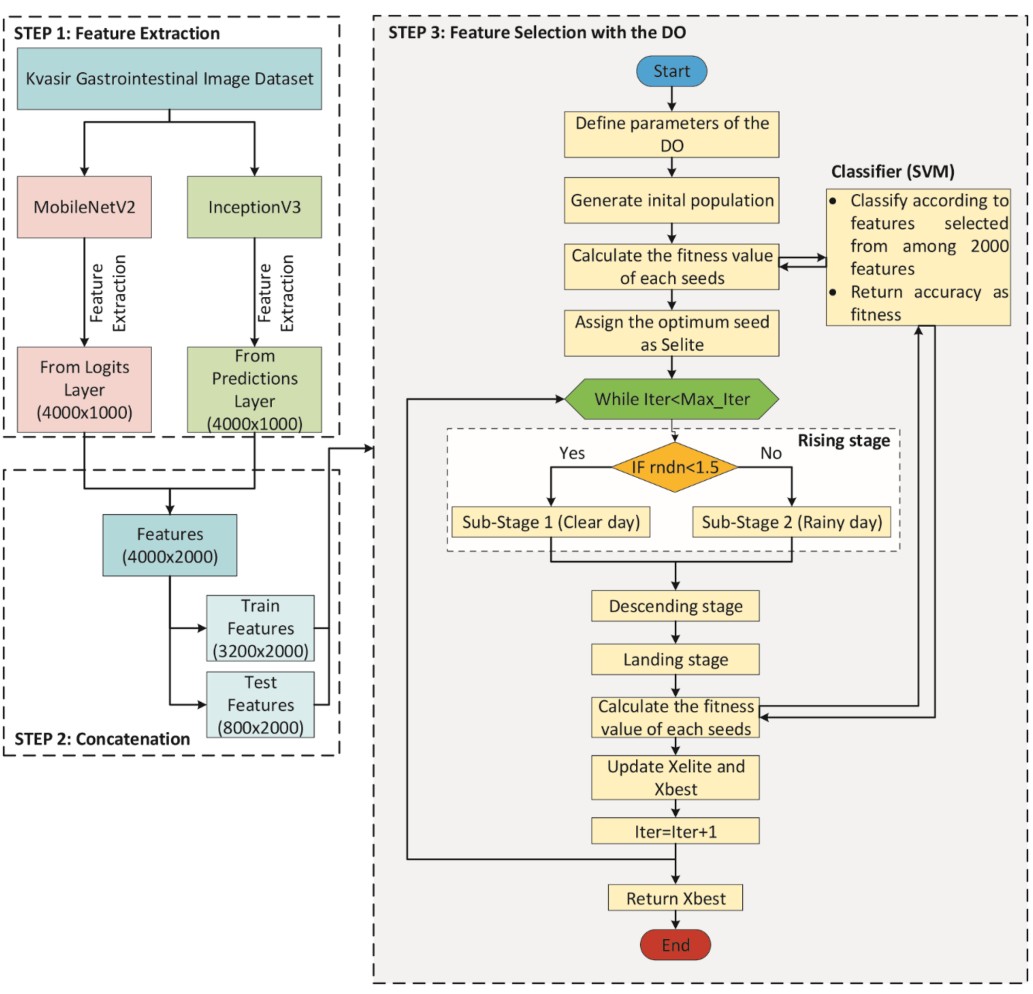

**Figure 3  Flowchart of DO-CNN architecture.**   

MobileNetV2 (*Sandler et al., 2018*), ShuffleNet (*Zhang et al., 2018*), and GoogleNet (*Szegedy et al., 2015*) pre-trained models, which are accepted in the literature, were used. The DO-CNN for classification of gastrointestinal diseases consists of three main steps and its flowchart is presented in Fig. 3.

In STEP 1, MobileNetV2 and InceptionV3 CNN models are used to extract features from the images in the dataset. For each image out of 4,000 images in the dataset, 1,000 features are extracted from both CNN models. This gives a 4,000 × 1,000 matrix from each of the two CNN models separately. Then, in STEP 2, the features obtained from these two CNN models are combined into a single vector. As a result of this process, a 4,000 × 2,000 feature matrix is obtained, in other words, 2,000 features for each image in the dataset. In addition, since the dataset is divided 80% for train and 20% for test in this step, feature matrices of 3,200 × 2,000 for train and 800 × 2,000 for test are obtained. Finally, in STEP 3, the feature selection process starts with the DO algorithm on these features. In the initial population of the DO, there are $N$ seeds (candidate solutions) of size 2,000, each with randomly generated values between 0 and 1. If the value of a dimension in a candidate

| Feature No: | 1 | 2 | 3 | 4 | 5 | 6 | 7 | ... | 2000 |
|---|---|---|---|---|---|---|---|---|---|
| Candidate solution: | 0.74 | 0.12 | 0.28 | 0.56 | 0.78 | 0.35 | 0.87 | ... | 0.62 |

**Figure 4 Encoding of a candidate solution.**

**Table 1 Performance evaluation metrics.**

| | |
|---|---|
| Accuracy = (TP + TN)/(TP+ TN+FP+FN) | F1-Score = 2TP/(2TP + FP + FN) |
| Sensitivity = TP/(TP + FN) | FPR = FP/(FP + TN) |
| Specificity = TN/(FP + TN) | FDR = FP/(FP + TP) |
| Negative predictive value (NPV) = TN/(TN + FN) | FNR = FN/(FN + TP) |

solution is less than 0.5, the feature will not be selected, and if it is greater than 0.5, the feature will be selected. For clarity, the encoding of a candidate solution is given in Fig. 4. As can be seen in Fig. 4, since there are 2,000 features for each image, a candidate solution consists of 2,000 dimensions with random values between 0 and 1. In the figure, the values in the 1st, 4th, 5th, 7th…2,000th dimensions of the candidate solution are greater than 0.5, so these features will be selected.

Then, the fitness value of each candidate solution in the DO is calculated using the SVM classifier. The classifier takes into account features corresponding to dimensions greater than 0.5 in the candidate solution for training and prediction. The accuracy value obtained at the end of this process is considered the fitness value of the candidate solution. Finally, the candidate solution with the highest fitness value is considered the problem solution.

## RESULTS

The MATLAB 2023 environment was used to achieve the study's results. A Windows operating system machine with an i7 processor and 16 GB of RAM was used to capture the results. While obtaining experimental results, the default parameters of the CNN architectures were used in the feature extraction process. In the feature selection phase, the parameters of the DO are set as follows: the initial population size is 50, the maximum number of iterations is 100, the lower and upper bounds of the candidate solutions are 0 and 1, respectively, $\beta$ is 1.5, and the adaptive parameters α and k vary between 0 and 1. The DO-CNN's performance was evaluated using various measures for performance measurement (*Yildirim, 2022*). The performance measurement metrics used are presented in Table 1.

In this study, out of 4,000 images in the dataset, 3,200 images were used for training and 800 images were used for testing. The results obtained using the test images are presented in the study.

In the confusion matrix; 1, 2, 3, 4, 5, 6, 7, and 8 represent Dyed and Lifted Polyps, Dyed Resection Margins, Esophagitis, Normal cecum, Normal pylorus, Normal Z-line, Polyps, and Ulcerative colitis, respectively.
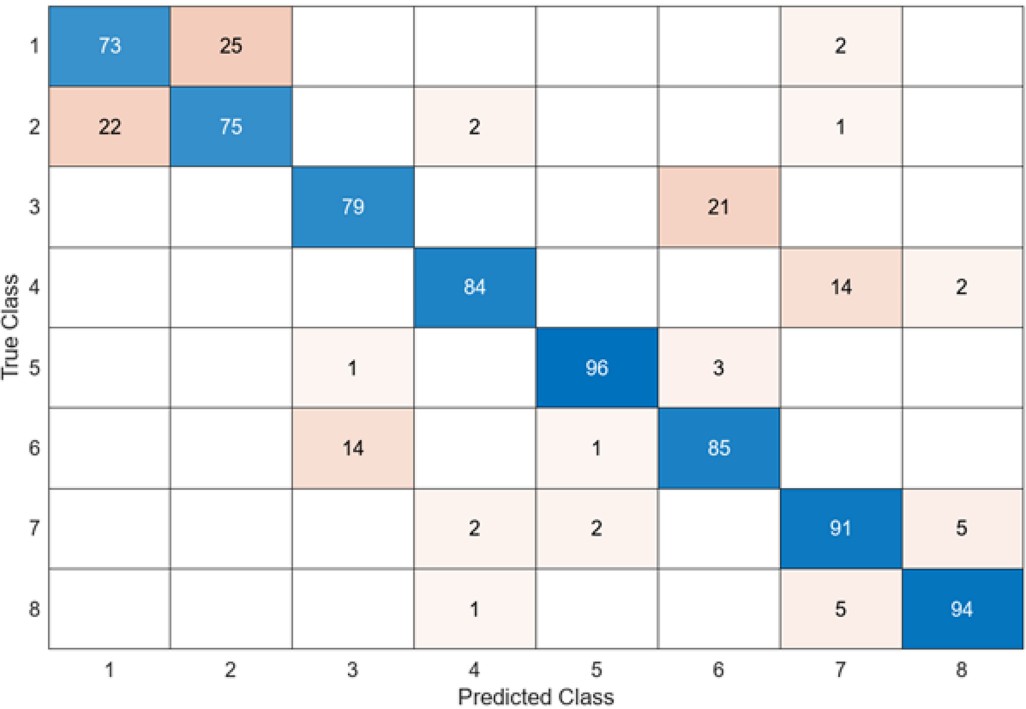

**Figure 5 GoogleNet + SVM.**                

## Classification of features extracted from pre-trained models in different classifiers

To compare the performance of the DO-CNN, feature extraction was executed using six different pre-trained models accepted in the literature. The dataset consists of 4,000 images in total and the number of features extracted in each architecture is 1,000. As a result, a 4,000 × 1,000 feature map was obtained. These obtained features were classified in the SVM classifier used in DO-CNN. While 3,200 of the 4,000 images in the data set were used for training the model, 800 images were used for testing the model. The confusion matrix obtained when the feature maps acquired using GoogleNet that are classified in the SVM is presented in Fig. 5.

When the confusion matrix obtained after training and testing the SVM classifier with the feature map obtained using the GoogleNet architecture is examined, it is seen that the accuracy value of the GoogleNet architecture is 84.6%. While the most successful class of the GoogleNet + SVM model is class 5, the least successful class is class 1. While the GoogleNet + SVM model correctly predicted 677 images out of 800 test images, it misclassified 143.

The confusion matrix obtained when the feature maps acquired using the AlexNet are classified in the SVM is presented in Fig. 6.

When the confusion matrix obtained after training and testing the SVM classifier with the feature map obtained using the AlexNet architecture is examined, it is seen that the accuracy value of the AlexNet architecture is 85.4%. While the most successful class of the

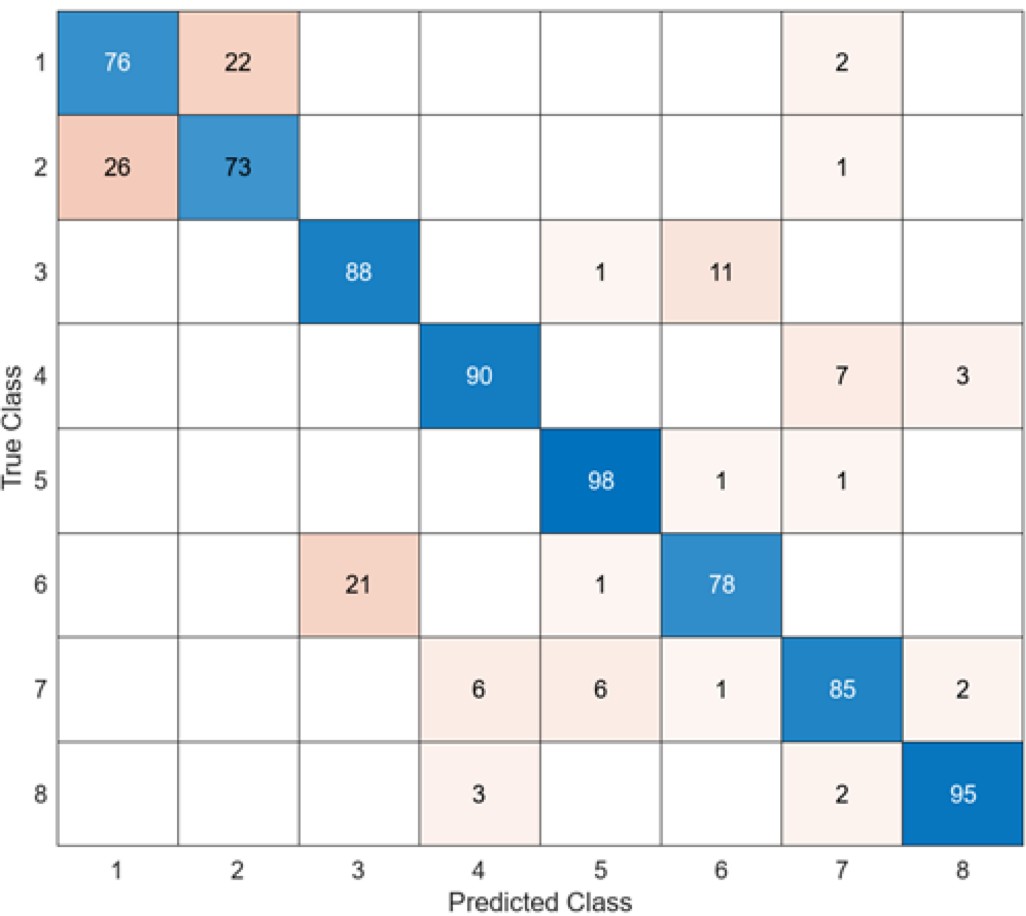

**Figure 6 AlexNet + SVM.**

AlexNet + SVM model is class 5, the least successful class is class 2. While the accuracy value achieved by the AlexNet architecture in class 5 was 98%, the accuracy value in class 2, where it was least successful, was 73%. While the AlexNet + SVM model correctly predicted 683 images out of 800 test images, it misclassified 137.

The confusion matrix obtained when the feature maps acquired using the InceptionV3 are classified in the SVM is presented in Fig. 7.

When the confusion matrix obtained after training and testing the SVM classifier with the feature map obtained using the InceptionV3 architecture is examined, it is seen that the accuracy value of the InceptionV3 architecture is 87.2%. While the most successful class of the InceptionV3 + SVM model is class 5, the least successful class is class 2. While the accuracy value of the InceptionV3 architecture in class 5 was 98%, the accuracy value in class 2, where it was the least successful, was 73%. While the InceptionV3 + SVM model correctly predicted 683 images out of 800 endoscopy test images, it misclassified 137 endoscopy images.

The test confusion matrix obtained when MobileNetV2 is used as a base and the resulting feature map is classified in the SVM classifier is shown in Fig. 8.

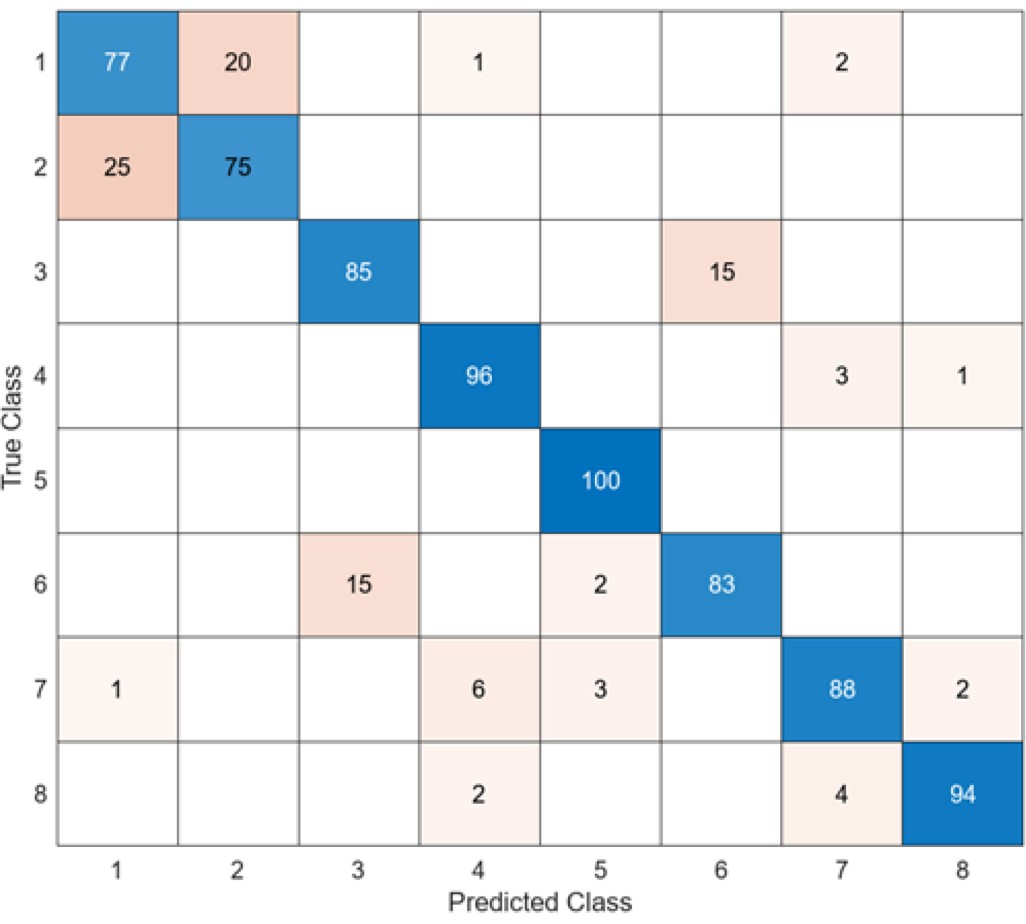

**Figure 7** **InceptionV3 + SVM.**

When the confusion matrix obtained after training and testing the SVM classifier with the feature map obtained using the MobileNetV2 architecture is examined, it is seen that the accuracy value of the MobileNetV2 architecture is 90.0%. While the most successful class of the MobileNetV2 + SVM model is class 5, the least successful class is class 1. While the accuracy value achieved by the MobileNetV2 architecture in class 5 was 97%, the accuracy value in class 1, which was least successful, was 82%. MobileNetV2 + SVM model predicted 720 images correctly and 80 endoscopy test images incorrectly out of a total of 800 endoscopy test images.

The test confusion matrix obtained when ResNet50 is used as a base and the resulting feature map is classified in the SVM classifier is shown in Fig. 9.

When Fig. 9 is examined, it can be seen that the accuracy value of the ResNet50 architecture is 89.5%. While the most successful class of the ResNet50 + SVM model is class 5 with an accuracy rate of 100%, the least successful class is class 1 with 73%. Out of a total of 800 endoscopy test images, the ResNet50 + SVM model predicted 716 images correctly and 84 endoscopy test images incorrectly.

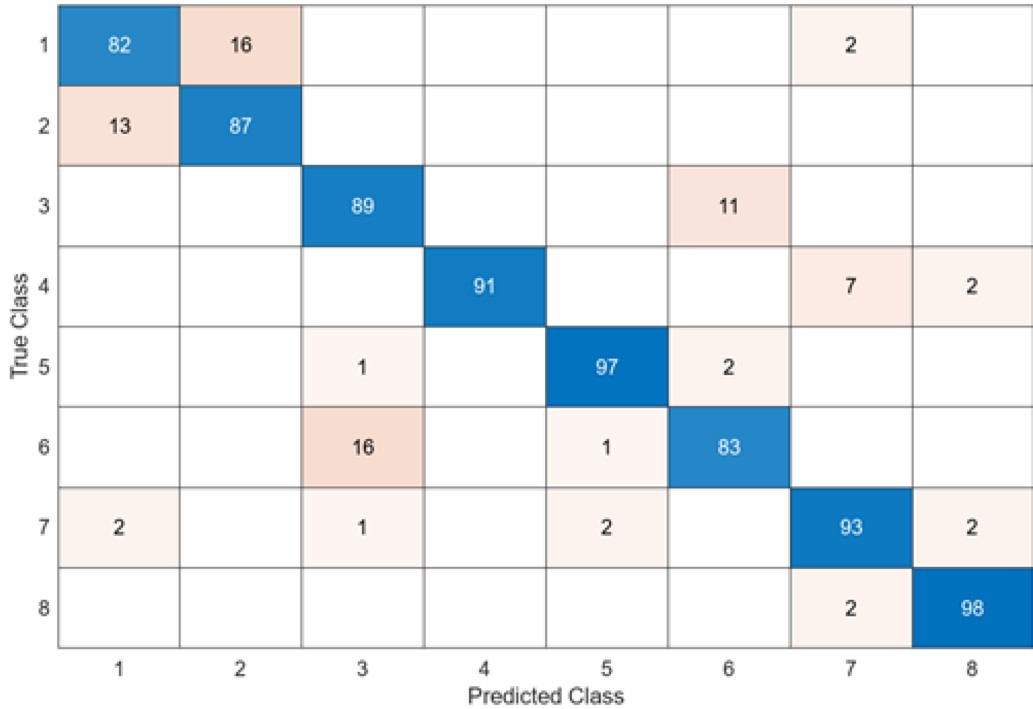

**Figure 8 MobileNetV2 + SVM.**

The last architecture used to compare the model proposed in the study is ShuffleNet. The confusion matrix obtained when the feature map obtained in the ShuffleNet architecture is classified in SVM is given in Fig. 10.

When Fig. 10 is examined, it can be seen that the accuracy value of the ShuffleNet architecture is 89.1%. While the most successful class of the ShuffleNet + SVM model is class 5 with an accuracy rate of 100%, the least successful class is class 2 with 80%. In class 2, 20 test images were predicted incorrectly. ShuffleNet + SVM model predicted 713 images correctly and 87 endoscopy test images incorrectly out of a total of 800 endoscopy test images.

After feature extraction was performed with six different architectures to compare the DO-CNN, the obtained features were classified in the SVM classifier. When six different confusion matrices are examined, it is seen that the error rate is highest in classes 1 and 2. This is because the images in classes 1 and 2 are very similar to each other. To solve this problem, models need to be supported with more data.

As a result, accuracy values of 84.6% in the GoogleNet + SVM model, 85.4% in the AlexNet + SVM model, 87.2% in the InceptionV3 + SVM model, 89.1% in the ShuffleNet + SVM model, 89.50% in the ResNet50 + SVM model and 90.0% in the MobileNetV2 + SVM model were obtained. The lowest accuracy value was reached at 84.6% in the GoogleNet + SVM model, and the highest accuracy value at 90.0% was achieved in the MobileNetV2 + SVM model.

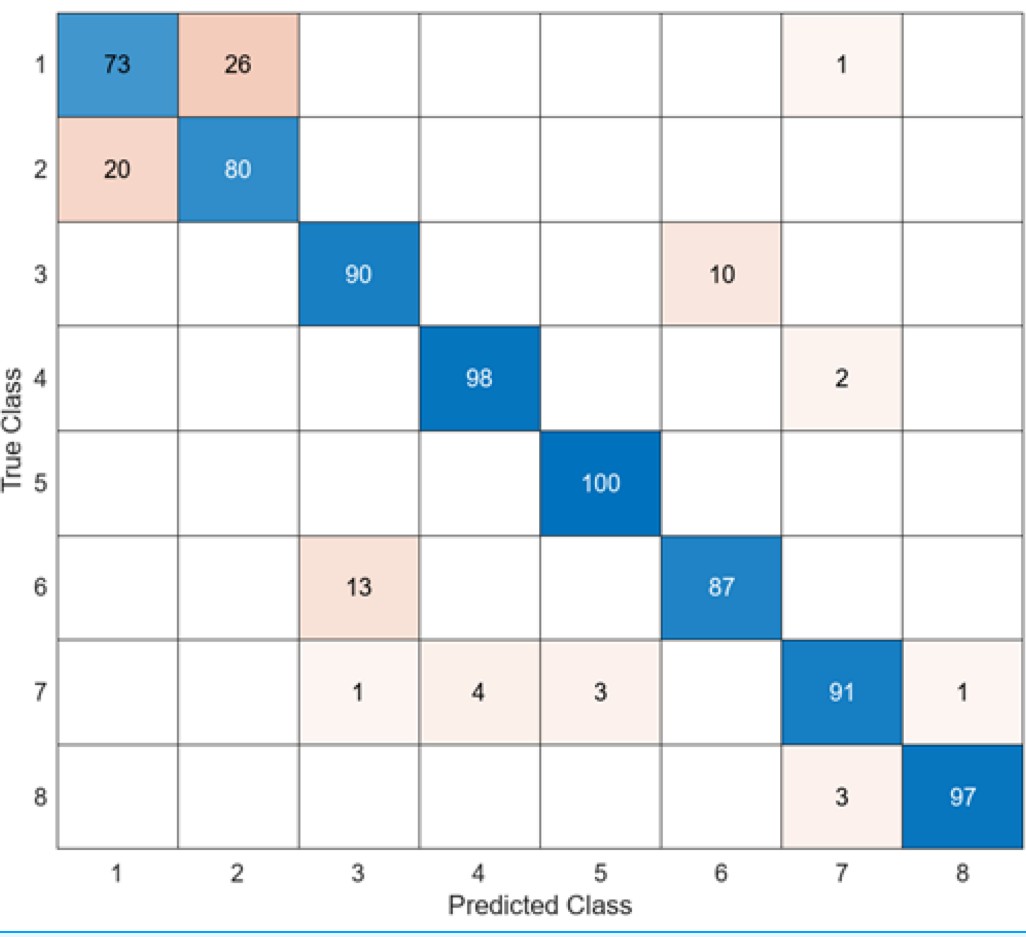

**Figure 9 ResNet50 + SVM.**

## Results obtained in the DO-CNN

In this section, the results of the proposed hybrid model for the classification of gastrointestinal diseases are examined. Figure 11 shows the confusion matrix of DO-CNN's best result obtained in 20 independent attempts.

Figure 11 reveals that the highest accuracy achieved by DO-CNN in 20 independent attempts was 93.88%. Class 5 has the highest accuracy rate of 100%, making it the most successful class in the DO-CNN model; class 1 has the lowest accuracy rate, at 84%. From a total of 800 endoscopy test images, the DO-CNN model predicted 741 images correctly and 59 endoscopy test images incorrectly. The mean, maximum and standard deviation values obtained by the DO-CNN architecture in 20 independent attempts are shown in Table 2.

The average accuracy value obtained with the proposed DO-CNN model in 20 independent trials is 92.9125%, the maximum accuracy value is 93.8750% and the standard deviation value is 0.3763. The performance measurement metrics of the DO-CNN model proposed for the classification of gastrointestinal diseases are presented in Table 3.

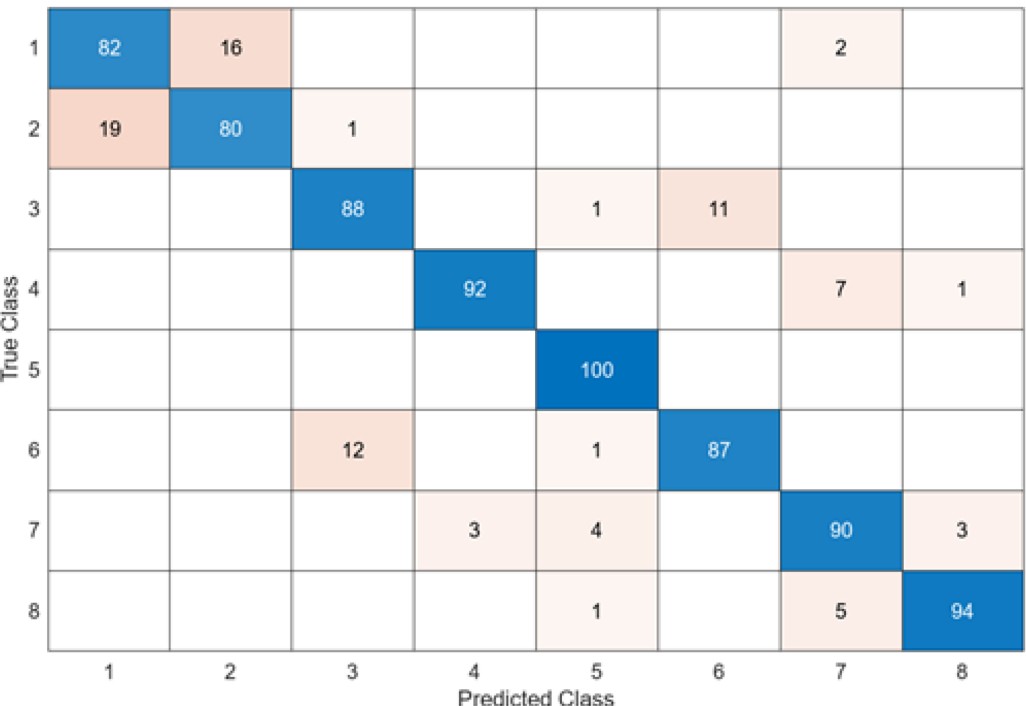

**Figure 10 ShuffleNet + SVM.**

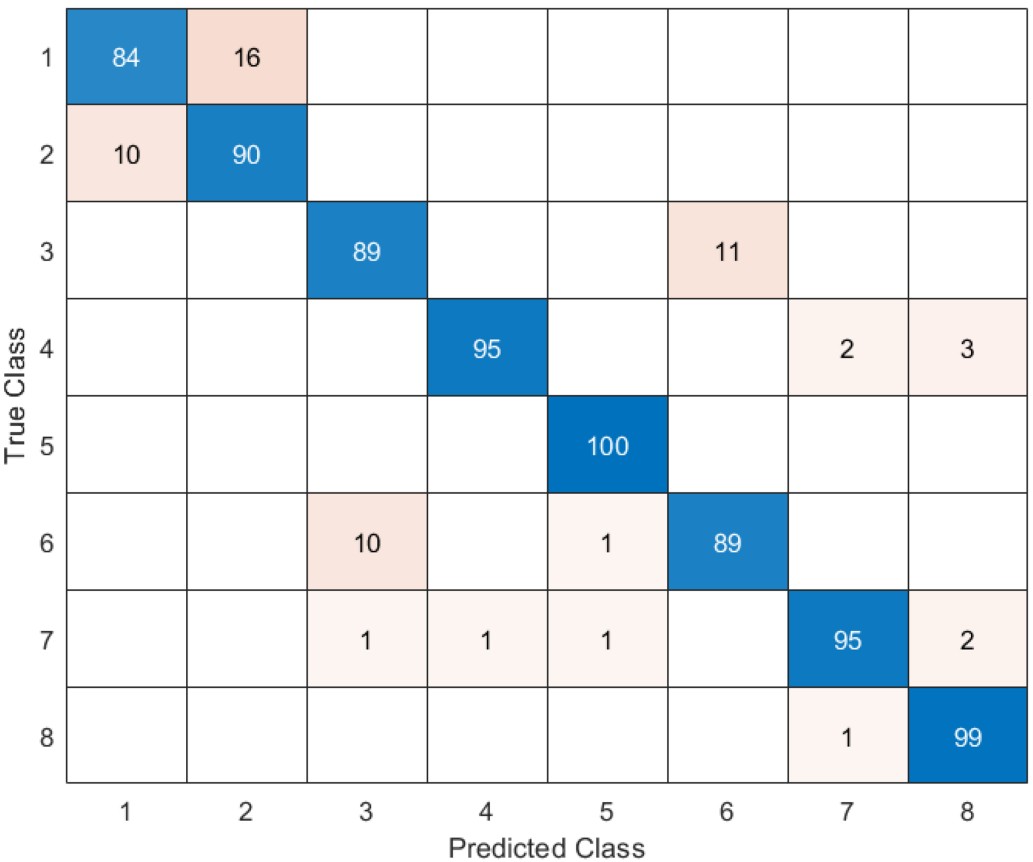

**Figure 11 DO-CNN's confusion matrix.**

**Table 2 Results of the DO-CNN model.**

|  | Mean accuracy | Max accuracy | Std. Dev. |
|---|---|---|---|
| DO-CNN (Proposed method) | 92.9125% | 93.8750% | 0.3763 |

**Table 3 Performance measurement metrics of the DO-CNN model (%).**

| Classes | Accuracy | Sensitivity | Specificity | NPV | F1-score | FPR | FDR | FNR |
|---|---|---|---|---|---|---|---|---|
| 1 | 84 | 89.36 | 97.73 | 98.57 | 86.60 | 2.27 | 16 | 10.64 |
| 2 | 90 | 84.91 | 98.56 | 97.71 | 87.38 | 1.44 | 10 | 15.09 |
| 3 | 89 | 89 | 98.43 | 98.43 | 89 | 1.57 | 11 | 11 |
| 4 | 95 | 98.96 | 99.29 | 99.86 | 96.94 | 0.71 | 5 | 1.04 |
| 5 | 100 | 98.04 | 100 | 99.71 | 99.01 | 0 | 0 | 1.96 |
| 6 | 89 | 89 | 98.43 | 98.43 | 89 | 1.57 | 11 | 11 |
| 7 | 95 | 96.94 | 99.29 | 99.57 | 95.96 | 0.71 | 5 | 3.06 |
| 8 | 99 | 95.19 | 99.86 | 99.29 | 97.06 | 0.14 | 1 | 4.81 |

Table 3 shows that the highest accuracy value is class 5 with 100%, followed by class 8 with 99%, classes 4 and 7 with 95%, class 2 with 90%, classes 3 and 6 with 89%, and class 1 with 84%.

## DISCUSSION

Gastrointestinal diseases are defined as disorders that can occur in any part of the digestive system. These diseases usually affect the organs of the digestive system: the mouth, esophagus, stomach, intestines, liver, pancreas, and gallbladder. Important gastrointestinal diseases include reflux, ulcers, irritable bowel syndrome, constipation, Crohn's syndrome, colon polyps, liver failure, gallbladder disease and pancreatitis. Patients should be examined by a gastroenterologist for diagnosis and treatment (*Davies, Black & Fairbrass, 2022*; *Peery et al., 2019*).

Several different methods are used to detect digestive system diseases. Physical examination: The doctor listens the problems with the gastrointestinal tract. Doctor then examines the abdominal area and evaluate for symptoms, pain, or abnormalities. Blood tests: Many gastrointestinal diseases can be diagnosed with blood tests. Blood tests include examining enzyme levels, blood cells, or markers of infection (*Odze & Goldblum, 2014*; *Gikas & Triantafillidis, 2014*).

Imaging methods: Images of the gastrointestinal system can be obtained using methods such as x-ray, ultrasound, computed tomography and magnetic resonance imaging. These methods are used to evaluate the structure, size, shape and abnormal structures of organs. Endoscopy: In this procedure, an endoscope shaped like a flexible tube is used. The endoscope is passed through the stomach or intestine, allowing visual inspection of the internal structures. It can also be used for biopsy. Colonoscopy: This is a type of endoscopy used specifically to detect intestinal diseases. In this procedure, an endoscope is inserted

**Table 4 Literature list of gastrointestinal diseases.**

| Article | Year | Method | Image number of dataset | Accuracy (%) |
|---|---|---|---|---|
| *Lonseko et al. (2021)* | 2021 | CNN | 12,147 | 93.19 |
| *Nouman Noor et al. (2023)* | 2023 | MobileNetV2 | 30,000 | 96.40 |
| *Mohapatra et al. (2021)* | 2021 | 2DDWT+CNN | 28,800 | 97.25 |
| *Varalaxmi et al. (2023)* | 2023 | ResNet50 | —————— | 88.05 |
| *Montalbo (2022)* | 2022 | CNN | —————— | 97.25 |
| Proposed model | 2023 | DO-CNN | 4,000 | 93.88 |

into the large intestine and the doctor evaluates the condition of the internal structures or performs a biopsy (*Scholz et al., 2015*; *Leighton et al., 2006*).

Stool tests: Some infections or digestive disorders can be seen in the stool. Gastrointestinal diseases can be diagnosed by examining a stool sample (*Garner et al., 2007*). Diagnosis of gastrointestinal diseases is usually made by a combination of one or more methods. To confirm the diagnosis and determine the appropriate treatment method, doctors make an evaluation by combining test results and patient history (*Seyedian, Nokhostin & Malamir, 2019*). Artificial intelligence techniques and metaheuristic optimization methods have been used extensively in recent years to help experts make the correct diagnosis (*Bacanin et al., 2023*; *Seyyedabbasi, 2023*). In this study, the problem of classification of gastrointestinal diseases was considered as a real-world engineering problem and the classification process was carried out for the first time with a hybrid method proposed in this article. The proposed method in this article for the classification of gastrointestinal diseases and similar studies in the literature, the year in which the studies were carried out, the method used, the number of images in the dataset and the performance of the method are shown in Table 4.

As can be seen in Table 4, there are studies that obtain higher accuracy values than the hybrid method proposed in this article. However, data augmentation techniques are not considered appropriate because they may cause overlearning of the models.

The limitations of this study can be summarized as follows. Its most important limitation is that disease images are collected from people in a certain region. While hereditary characteristics may vary from region to region, they may also be reflected in disease images. It is expected that there will be much more image data in a dataset containing eight different classes for the purpose of detecting and classifying gastrointestinal diseases. Adjusting the optimized parameters of the combined algorithms seems another limitation of the proposed hybrid method. Furthermore, combined methods in these study are iterative and stochastic in nature and that is why, these methods can produce different solutions in different runs.

In future studies, it is planned to create a dataset containing gastrointestinal disease images obtained from multiple centers. In addition, it is planned to increase the number of images in the dataset to be created as much as possible.

## CONCLUSIONS

This study on the classification of gastrointestinal diseases with artificial intelligence offers a potential research area. With the increasing use of artificial intelligence technology in the field of health, more studies need to be done and different artificial intelligence models need to be developed. The results of this study showed that the diagnosis and classification processes of gastrointestinal diseases can be carried out more effectively with artificial intelligence. In this study, gastrointestinal diseases were tried to be classified with a new hybrid model created by using a metaheuristic optimization algorithm inspired by nature and convolutional neural networks. During the study, original data were used and data augmentation methods were not followed. The DO-CNN model can be used as a tool to assist doctors in the diagnosis process of gastrointestinal diseases. The use of artificial intelligence models, especially in the diagnosis of diseases that require expertise and have high complexity, can provide accurate and rapid detection. This can offer patients a more effective treatment process.

## ACKNOWLEDGEMENTS

The authors thank the dataset owners for sharing their datasets.

### Funding

The authors received no funding for this work.

### Competing Interests

Bilal Alatas is an Academic Editor for PeerJ.

### Author Contributions

- Soner Kiziloluk conceived and designed the experiments, performed the experiments, performed the computation work, prepared figures and/or tables, and approved the final draft.
- Muhammed Yildirim conceived and designed the experiments, performed the experiments, analyzed the data, performed the computation work, prepared figures and/or tables, and approved the final draft.
- Harun Bingol conceived and designed the experiments, performed the experiments, analyzed the data, prepared figures and/or tables, authored or reviewed drafts of the article, and approved the final draft.
- Bilal Alatas conceived and designed the experiments, analyzed the data, authored or reviewed drafts of the article, and approved the final draft.

### Data Availability

The raw data are publicly available at Kvasir: https://datasets.simula.no/kvasir/. The source code is available in the Supplemental File.

## Supplemental Information

Supplemental information for this article can be found online at http://dx.doi.org/10.7717/peerj-cs.1919#supplemental-information.

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
