# Peer review of "Multi-feature fusion and dandelion optimizer based model for automatically diagnosing the gastrointestinal diseases"

_PeerJ Computer Science, doi:10.7717/peerj-cs.1919_

## Round 0.1 · original submission · Major Revisions

Dear authors

Thanks for your submission. Based on the input from the experts, you need to incorporate suggested improvements.

Please clearly state the contribution of your work as well.

·

Basic reporting

Why specifically dandelion optimizer algorithm is adapted for the problem of diagnosing the gastrointestinal diseases is not clear.
Although there are many global optimization methods, the authors should write the reason and motivation of this selection.
There are many personal pronouns (such as first-person usage) in the article. This approach should be avoided in academic writing because in truly scientific writing, personal identity should play no role in the validity or framing of the research and reporting.
Why the features of Kvasi gastrointestinal image data are firstly concatenated (increased) and then selected (decreased) is not clear.
Many of the hybridized iterative methods need parameters that should be adjusted a priori. How they are set is not clear.
Configuration space dandelion optimizer algorithm is not clear. It should be more specific and comprehensive. Proper graphical expression to explain the proposal should be given. Each process in the proposal should be clearly described. Encoding type of the dandelion optimizer algorithm for feature selection problem should be detailed. It can also be shown as figure. Decision variables, their types, their boundaries, etc. should be shown. If normalization for the variables is performed for the dandelion optimizer, this should be written within the text. How the binarization/categorization scheme and approach are followed is not presented. Is dandelion optimizer a single model for diagnosing the gastrointestinal diseases or it is used as feature selection for a classification algorithm is not clear in the paper.
The values of the parameters of all methods should be clearly stated.

Experimental design

How the constraints of the focused problem are coped with by the adapted dandelion optimizer is not clear. Are transfer functions are used? If this is the case, which are used? If not, how this problem is handled.
The starting points of the algorithms are not clear. Are they identical for all algorithms? Have the simulations been done in the same situations? How can a fair comparison be guaranteed?
Importance and contribution are not highlighted. Specific contribution to knowledge should be clearly stated.
What are the limitations or constraints of the methodology adopted in this work? What are the other possible methodologies that can be used to achieve this objective in relation to this diagnosing the gastrointestinal diseases work?

Validity of the findings

How the constraints of the focused problem are coped with by the adapted dandelion optimizer is not clear. Are transfer functions are used? If this is the case, which are used? If not, how this problem is handled.
The starting points of the algorithms are not clear. Are they identical for all algorithms? Have the simulations been done in the same situations? How can a fair comparison be guaranteed?
Importance and contribution are not highlighted. Specific contribution to knowledge should be clearly stated.
What are the limitations or constraints of the methodology adopted in this work? What are the other possible methodologies that can be used to achieve this objective in relation to this diagnosing the gastrointestinal diseases work?

Additional comments

References should be written correctly according to the style of the PeerJ Computer Science journal reference.
Use “metaheuristic” or “meta-heuristic” but not both. All of them can be corrected as “metaheuristic”.
Variables should be written in italics, similar to their use in equations.
In Figure 2, why the random values is compared to 1.5 is not clear.
What does “T” in Figure 2 represent in not mentioned. The termination criteria of dandelion optimizer algorithm in the experiment is not clear. If this T is maximum iteration number, it should be mentioned in the “Application Results” section.
Attention should be paid to the use of the space character.
Equations must be used with the correct equation numbers.

Reviewer 2 ·

Basic reporting

I examined your article titled "Multi-feature fusion and dandelion optimizer based model for automatically diagnosing the gastrointestinal diseases" in detail. I would like to point out that the article is generally well-written. However, there are points in the article that are missing and need to be clarified. I have explained these respectively below:
1- In the Abstract section, it is mentioned that a high accuracy value has been achieved in line 49. This accuracy value needs to be added to the relevant section.
2- I think the Contribution and Novelty section is presented in great detail. The relevant section should be revised to avoid repetitions. Additionally, the importance of feature concatenation should also be included in this section. It should be clearly written why the feature concatenation step is applied.
3- Is the use of InceptionV3 and MobileNetV2 architectures as a basis in the proposed model random, or is there a point on which it is based?
4- The motivation for using support vector machines as base shallow learning methods should be written.
Only rough subheadings were written in the "Organization of Paper" section. Adding brief descriptive information about subheadings in this section will increase the reader's interest in the article.

Experimental design

5- This reviewer thinks whether a deep network is required depending on the focused problem considering the volume of data? A better justification for the use of the deep network in this work is needed (although the methodology seems original and promising results are obtained from the proposed hybrid model).
6- Training and testing steps of the proposed optimization-based deep architecture model should be presented in detail. The use of testing methods requires further motivation and explanation. Why is cross-validation not used?

Validity of the findings

7- It is not clear how the mean accuracy value is obtained from the DO-CNN model as 92.9125 is not explained.
8- Limitations are not discussed in detail. Constraints and disadvantages concerning the proposed hybrid model should also be presented.
9- An additional explanation is required for using the dandelion optimization algorithm for the related feature selection process.
10- There may be many different local minima in the cost function during the weight optimization process of deep models, which is why deep networks behave in a somewhat random way. The authors should write a few sentences about the results according to this randomness.
11- In this work, a number of selected features from the images are given as input to the deep networks. The weighting factors can be adjusted during the training phase in the proposed model, however if there are more or different given inputs, the weighting factors could be different. Would this affect the conclusion? How do you deal with this problem? The generality and robustness properties of the proposed method may be discussed.
12- It is unclear why a specific intelligent optimization method is tailored to the task of identifying gastrointestinal disorders from images. Even though there are numerous global search and optimization methods, the authors may explain why and how they came to make this choice.
13- Values of the parameters for the proposed optimization method should be listed in the paper.
14- It is not clear how relevant and important features are selected by the optimization algorithm. Selecting or deselecting is a binary work, however this optimization method works on numerical values. Binarization should be clearly explained.
15- The risk of overfitting in deep models is high due to the high number of parameters. Has overfitting been encountered?

Additional comments

16- “…computed tomography (CT)” should be written as only “computed tomography” because there is no used “CT” in the paper. This is not an abbreviation in this paper.
17- “…magnetic resonance imaging (MRI).” should be written as only “magnetic resonance imaging” without an abbreviation because there is no usage as MRI in the paper.
18- Equations are part of the sentence and that is why special attention should be paid in these sentences.
Finally, it is important to review the spelling and grammatical errors in the paper.

---

## Round 0.2 · accepted · Accept

Dear authors,

thank you for your re-submission. Based on the experts' opinions and my judgment, I am pleased to inform you that your manuscript has been recommended for publication.

Thank you for your contribution.

·

Basic reporting

The methodology employed in this research is robust and well-executed, ensuring the reliability and validity of the results

Experimental design

In the revised version, authors improved experimental results and presentation of main contributions.

Validity of the findings

The methodology employed in this research is robust and well-executed, ensuring the reliability and validity of the results

Reviewer 2 ·

Basic reporting

All changes have been completed.

Experimental design

All changes have been completed.

Validity of the findings

All changes have been completed.